# Fungal Allergen and Mold Allergy Diagnosis: Role and Relevance of *Alternaria alternata* Alt a 1 Protein Family

**DOI:** 10.3390/jof8030277

**Published:** 2022-03-09

**Authors:** Patricia Sánchez, Ainara Vélez-del-Burgo, Ester Suñén, Jorge Martínez, Idoia Postigo

**Affiliations:** Department of Immunology, Microbiology and Parasitology, Faculty of Pharmacy and Laboratory of Parasitology and Immunoallergy, Lascaray Research Centre, University of the Basque Country, 01006 Vitoria-Gasteiz, Spain; patricia.sanchez@ehu.eus (P.S.); ainara.velezdelburgo@ehu.eus (A.V.-d.-B.); ester.sunen@ehu.eus (E.S.); jorge.martinez@ehu.eus (J.M.)

**Keywords:** Alt a 1, fungal allergy, asthma, *Alternaria*, allergic disease

## Abstract

*Alternaria* is a genus of worldwide fungi found in different habitats such as soil, the atmosphere, plants or indoor environments. *Alternaria* species are saprobic—largely involved in the decomposition of organic material—but they can also act as animal pathogens, causing disease in humans and animals, developing infections, toxicosis and allergic diseases. *A. alternata* is considered one of the most important sources of fungal allergens worldwide and it is associated with severe asthma and respiratory status. Among the *A. alternata* allergens, Alt a 1 is the main sensitizing allergen and its usefulness in diagnosis and immunotherapy has been demonstrated. Alt a 1 seems to define a protein family that can be used to identify related pathogenic fungi in plants and fruits, and to establish taxonomic relationships between the different fungal divisions.

## 1. Introduction

Allergic diseases are a group of immune-mediated disorders with a high worldwide prevalence affecting 30% of the general population and increasing dramatically. Asthma, rhinitis; anaphylaxis; drug, food, and insect allergy; eczema; urticaria (hives) and angioedema are included in this group of diseases.

The main allergens causing type I allergies are contained in different sources, of which mites, pollens, epithelia, fungi and some foods are the most important [1,2]. Fungi are the fourth source of sensitization involved in allergic respiratory diseases preceded by pollens and dust mites. The exposure to allergenic molds can cause IgE-mediated allergic rhinitis, asthma and atopic dermatitis [3], and it has been described that the sensitization to some fungi such as *Aspergillus* and/or *Alternaria* is related to the severity of an asthmatic arrest [4].

There are several epidemiologic studies about fungal allergy prevalence, but the data vary according to the method of diagnosis [4]. It is estimated that the prevalence of allergies to molds is about 3–10%, although the prevalence of fungal sensitization shows a wide geographical variability [3]. Among the allergenic mold species, *Alternaria alternata*, *Aspergillus fumigatus*, *Cladosporium herbarum* and *Penicillium notatum* are the ones with the highest prevalence in allergic disease or sensitizations [5]. Data from the European Community show a prevalence of positive skin prick tests (SPT) using *Alternaria* and *Cladosporium* allergenic sources ranged from 0.2% to 14.4% in the adult general population (aged 20–44) [6]. In the USA, fungal sensitization of asthmatic patients revealed a prevalence of 80% [7]. *A. alternata* is considered one of the most important fungal allergens worldwide and is associated with severe asthma and respiratory status [4,8]. It accounts for 60% of SPT among patients sensitized to fungi [9].

*Alternaria* is a worldwide genus of Deuteromycetes fungi found in different habitats, such as soil, the atmosphere, plants or indoor environments, that include many saprophytic and pathogenic species. The taxonomy of these fungi has mainly been based on morphological conidial characters and to a lesser extent on host association, biochemistry and metabolites [10]. Recent phylogenetic studies have made significant changes to the systematic taxonomy within *Alternaria* by elevating 26 clades to the subgeneric taxonomic status of section [10]. According to phylogenetic and morphological studies, *Alternaria* contains most *Alternaria* species with conidia succession, including important plant, human and postharvest pathogens. In spite of the fact that the majority of *Alternaria* species are saprobic—largely involved in the decomposition of organic material—many of their species are endophytic, living in various parts of crops including leaves, seeds and fruits [11]. To a lesser degree, they can act as animal pathogens, causing disease in human and animals, developing infections, toxicosis and allergic diseases [10,12]. In the field of human ailments, allergic disease is undoubtedly the most common human pathology caused by *Alternaria* [11,13,14,15]. The Global Asthma and Allergy European Network (GA2LEN) is a consortium of leading European research centers specialized in allergic diseases, which include asthma. It was funded by the European Union under the 6th Framework Programme with the purpose of addressing the growing public health concern of allergic diseases. The GA2LEN initiative performed a study in 14 countries of the European Community (*n* = 3034) and showed a prevalence of sensitization to *A. alternata* of 9%, ranging from 2% in Finland to 23.8% in Greece [16]. In the United States, the prevalence of *A. alternata* sensitization in the general population (aged 6 to 74 years) was about 13% [7]. In Spain, this prevalence was estimated to be around 20% [17].

## 2. Fungal Allergen and Mold Allergy Diagnosis

The diagnosis and treatment of human allergic diseases classically involves the use of extracts obtained from biological material. However, in recent decades the application of molecular diagnostic procedures using individualized allergens has been decisive in the management of atopic individuals and allergic patients [18]. Described more than 20 years ago, it is known that the Alt a 1 allergen is the main allergen of *A. alternata* [19,20,21], with more than 80% of *Alternaria* allergic patients being sensitive to this allergen [3,22,23]. Since then, several other allergens belonging to this fungal species have been described (Table 1).

The official WHO/IUIS database currently lists 113 fungal allergens belonging to various protein families, of which 88 belong to Ascomycota species, 23 to Basidiomycota and 2 to Mucorales. The order Eurotiales includes 48 allergens belonging to *Aspergillus* and *Penicillium* species. The order Capnodiales includes 10 allergens belonging to *Cladosporium* species, and the Pleosporales order includes 18 allergens belonging to *Alternaria*, *Epicoccum*, *Ulocladium* and *Curvularia* species [24].

To date, only 10 individualized allergens from 5 mold species are available for allergy diagnosis and mold sensitization testing. These include nAlt a 1/rAlt a 1, the major allergen in *A. alternata*-sensitized individuals, and *A. alternata* enolase, rAlt a 6, with its potential cross-reactivity to molds and to natural latex allergens. The recombinant allergens rAsp f 1, 2, 3, 4 and 6 from *Aspergillus fumigatus* for allergy diagnostic purposes, including the specific evaluation of allergic bronchopulmonary aspergillosis (ABPA) and, finally the rCla h 8 allergen, considered the major allergen of *Cladosporium herbarum* with possible cross-reactivity to other dehydrogenase allergens [25].

The clinical and diagnostic relevance of individualized allergens and allergen extracts from seven common genera of fungi has been extensively reviewed [4,5]. Most authors agree on the importance of genera *Alternaria*, *Cladosporium*, *Penicillium*, *Aspergillus*, and *Malassezia* in clinical diagnosis [13,15,26,27,28,29]. The *Alternaria* and *Cladosporium* species are considered the most important outdoor allergens and the sensitization to species of these genera is associated with the development of asthma and rhinitis and exacerbation of asthma [13,15,26]. The *Penicillium* and *Aspergillus* species are both involved in allergic diseases as indoor allergens [27]. *A. fumigatus* is able to colonize the bronchial tract of asthmatic patients, causing acute persistent asthma and ABPA [28]. The *Malassezia* species are common commensals of healthy skin and may be associated with atopic dermatitis [29].

However, in the association between allergic diseases and fungal species, which have been identified in both indoor and outdoor environments, the fungal concentration and distribution of the species in both environments should be taken into account in any case [5]. Sharpe et al. cite that *Cladosporium*, *Alternaria*, *Aspergillus* and *Penicillium* species were found to be present in higher concentrations in homes of asthmatic participants [30]. The presence of high concentrations of spores belonging to these fungal genera increased the exacerbation of asthma symptoms compared to individuals exposed to lower concentrations of these fungi [30].

Rapid progress in gene and protein technology over the past decades has contributed significantly to the identification of species-specific and cross-reactive allergen molecules from different allergenic fungal sources [5]. Nevertheless, data verifying the clinical and diagnostic relevance of these allergens are insufficient and, more extensive studies are needed. Studies conducted in Japan in 2015, suggested the possible contribution of other fungal allergen sources, including Basidiomycete species (*Schizophyllum commune* and *Bjerkandera adusta*) to pathogenesis of respiratory diseases, including allergic asthma/rhinitis or allergic bronchopulmonary mycosis [5].

## 3. Relevance of Mold Distribution and Fungal Extracts in Allergy Diagnosis

The distribution and quantification of indoor and outdoor fungal species is strongly associated with the development of fungal allergy [27,31]. The mycological catalogs of fungal species that include aerobiological data are different according to region, climate and meteorological conditions of the geographical area studied [32,33]. The results reported in each country are clearly different and include a high number of genera belonging to Ascomycota, Basidiomycota, Mucoromycota and Deuteromycota [34,35,36,37,38]. Regarding the quantification of spores, only a few genera have maximums of more than 100 spores/m^3^, such as *Agaricus* (350 spores/m^3^); *Alternaria* (180 spores/m^3^); *Aspergillus*/*Penicillium* (450 spores/m^3^); *Coprinus* (500 spores/m^3^) and *Ustilago* (750 spores/m^3^); *Cladosporium* (4500 spores/m^3^) [28]. Most studies agree that *Cladosporium* is the genus that gives rise to the highest concentration of airborne spores [34,35,36,37,38,39,40]. Spores of other taxa considered the most important in allergy development corresponding to *Aspergillaceae* (*Aspergillus*, *Penicillium*) and *Pleosporaceae* (*Alternaria* section *Alternaria*, *Stemphyllium*, *Alternaria* section *Ulocladium*), also have significantly higher levels of spores in the air [31,34,35,36,37,38,39,40].

The prevalence of fungal sensitization shows a significant geographic variation with several fungal species involved in allergic diseases including the genera *Alternaria*, *Aspergillus*, *Aureobasidium*, *Bipolaris*, *Botrytis*, *Candida*, *Curvularia*, *Cladosporium*, *Drechslera*, *Epicoccum*, *Fusarium*, *Mucor*, *Penicillium*, *Phoma*, *Saccharomyces*, *Malassezia*, *Trichphyton*, *Stemphylium*, *Ganoderma*, *Ustilago* and *Pleurotus* [5,23,28,41], but *Alternaria*, *Aspergillus* and *Cladosporium* are the most reported sensitizing genera of fungal nature [3,41,42,43]. This would be in line with the most relevant fungal allergens found in indoor environments, e. g., Asp f 1 (Mitogillin), Alt a 1 (Unknown protein), Cla h 6 (Enolase), Cla h 8 (Mannitol dehydrogenase), Pen ch 13 (Serine protease) and Pen ch 18 (Serine protease) [44].

However, it must be considered that the quality of fungal allergenic extracts used as a diagnostic tool is not similar for all species due to several difficulties in the process of obtaining the extract such as fungal identification, culture conditions and standardization, strain variation, biological standardization, use of heterogeneous in-house reference sera and other biological, biochemical and immunochemical parameters [28,45,46]. These factors represent serious limitations to assert that the four genera indicated as the most relevant in allergy diagnosis are the only ones to be taken into account. Are there other species that, due to difficulties in obtaining diagnostic reagents, go unnoticed?

On the other hand, various fungi are known to cause sensitization in asthmatics, but the most important fungal agents causing severe asthma with fungal sensitization are not well defined [12]. According to data reported in the literature only *Alternaria*, *Aspergillus*, *Penicillium* and *Cladosporium* seem to be the strongest candidates for the development of severe asthma and rhinitis [4,13,36,47,48,49,50]. There are only a few case reports about the association between occupational fungal exposure and asthma involving the species: *Pleurotus cornucopiae*, *Lycopodium clavatum*, *Dictyostelium discoideum*, *Rhizopus nigricans* and *Neurospora* [13], whereas sensitization to *Alternaria, Cladosporium* and *Aspergillus* has a strong link with asthma severity [51,52,53,54]. Patients monosensitized to fungal allergens are associated with a high risk of developing clinical symptoms [55], while those who present multiple sensitizations to fungi are associated with poor asthma control [56].

## 4. Molecular Diagnosis and Fungal Molecules

The application of genomics and proteomics to the study of allergenic proteins has been the basis for building more accurate diagnostic tools. The identification and production of the individual allergen molecules has been key to addressing hitherto unresolved concepts, such as the causes of cross-reactivity phenomena, through molecular diagnosis.

At the beginning of the 21st century, Valenta et al. described the basis for the molecular diagnosis of allergy, a classical technique to measure specific IgE concentrations in serum by coupling individualized native or recombinant allergens to the solid phase (ImmunoCAP^®^) [18]. A few years later, the microarray/macroarray techniques were applied to the component resolved diagnosis and immunotherapy concept (CRD/CRIT). These techniques made available the wide IgE reactivity profiles of individuals to achieve a complete individualized management of the allergic patient [57].

Currently, the diagnostic accuracy and effectiveness of replacing allergenic sources by individual allergens is a fact. By applying the CRD, it is possible to minimize several problems with the standardization of allergenic extracts, such as fungal strains and batch-to-batch variability, the choice of fungal structure to be used as raw material, the type of culture and technology used to prepare allergenic extracts and the self-degradation of the extract once obtained [18].

Allergy molecular diagnostic is improving the routine care. Currently, more than 130 allergens from more than 50 sources are commercially available and their use represents an undeniable advance. This is especially noticeable in the case of food and insect allergy and for the selection of allergen immunotherapy [28,58].

Of the 113 defined individualized fungal allergens, only 8 are available to perform the molecular diagnosis of fungal allergy and define clinical patterns associated with fungal allergens. For example, data reported from the literature show that *Aspergillus* allergens could differentiate *Aspergillus* sensitization from ABPA [59,60,61].

### Alternaria Alternata Allergens in Molecular Diagnosis

Few works have been published addressing the diagnosis of fungal allergy using the CRD with the most common fungal allergens available [3,8,23,42,62,63,64]. All of the existing studies agree that *A. alternata* sensitization is the most prevalent in allergic patients to fungi and that Alt a 1 is an accurate marker to test species-specific sensitization or IgE-mediated allergy to *A. alternata.* Ninety percent of patients or more labeled as allergic to *Alternaria* react specifically to Alt a 1.

Postigo et al. reported that 28 out of 30 patients diagnosed with an allergy to *Alternaria* revealed specific IgE values to Alt a 1, and the other two were positive to Alt a 6, showing highly relevant specific IgE values [23]. The demonstration that Asp f 6 has a homologous allergen in *A. alternata* (Alt a 16) supports the mislabeling of *Alternaria* allergy and the possible diagnosis of *Aspergillus* allergy with connotations to ABPA [65]. Postigo et al. also demonstrated that 2 out of 30 patients revealed low levels of specific IgE to Alt a 1 and specific IgE to *Curvularia* was 4–6 times higher [23]. Considering that the *Curvularia* major allergen is a homolog of the Alt a 15, it could be considered the mislabeling of the *Alternaria* allergy and the possible diagnosis of an allergy to *Curvularia* [66]. These results suggest that to define the main sensitizer not only sensitizing the allergen profile is useful, but also the quantification of specific IgE to each positive allergen reaction. This is especially important in the management of the allergic patient, including the decision on immunotherapy. According to Rodriguez et al., specific IgE to Alt a 1 should be assessed before considering the prescription of AIT in a patient with a clinically relevant sensitization to *Alternaria* [53]. In these cases, the major allergen Alt a 1, should be recognized as a marker of primary sensitization to this fungus. Luengo et al. cite that Alt a 1 should be assessed when considering mold immunotherapy [67]. A recent clinical trial has demonstrated the efficacy and safety of allergen immunotherapy with a commercial extract of Alt a 1 [68,69].

## 5. Alt a 1 Protein Family Is a Phylogenetic-Related *Alternaria* Marker for Taxonomy, Allergy and Phytopathology

Although Alt a 1 has been a defined allergenic protein since 1991 [19,21], in 2012 it was defined and described as a unique β-barrel protein dimer found exclusively in fungi. The only allergens with a similar structure to Alt a 1 are lipocalins, which have an α-helix in addition to a β-barrel. Despite the similarities between lipocalins and Alt a 1, they are the homologs of Alt a 1 which define a distinct structural family of proteins [70].

Current phylogenetic studies have made significant changes to the systematic taxonomy of *Alternaria* by elevating 26 clades to the subgeneric taxonomic status of section [10]. *Alternaria* section consists of only 11 phylogenetic species and 1 species complex [71,72,73].

Alt a 1 gene sequences contain more parsimony-informative sites than other phylogenetic markers. Analyses of Alt a 1 gene strongly support the clustering of *Alternaria* spp. and related taxa into several species-groups: infectoria, alternata, porri, brassicicola, sonchi, radicina and embellisia group. The monophyly of the *Nimbya* group was moderately supported and the monophyly of the *Ulocladium* group was weakly supported [74]. Despite the high levels of variation in amino acid sequences, the results of in silico prediction of protein secondary structure for Alt a 1, demonstrated a high degree of structural similarity between most of the species, suggesting conservation of function [10,12,74].

According to this concept, several authors have used Alt a 1 DNA sequence or Alt a 1 protein to identify and quantify *A. alternata* from the environment, this being of particular interest for associating *Alternaria* with sensitization or respiratory allergy [55,75,76]. Gabriel et al. were able to identify Alt a 1 homologs from *A. alternata*, *A. tenuissima*, *A. infectoria*, *U. botrytis* and *S. botryosum* using different Alt a 1 expression gene sequences. The specific sequence of Alt a 1 was able to detect an amplicon of approximately 390 bp from Alt a 1, encoding genes from species closely related taxonomically to *A. alternata*, such as *A. tenuissima*. By contrast, the PCR system using a conserved sequence of Alt a 1 homologs was able to detect an amplicon of approximately 180 bp from Alt a 1 and Alt a 1-like encoding genes from *A. alternata*, *A. tenuissima*, *A. infectoria*, *U. botrytis* and *S. botryosum* [77]. Similar results were reported by Teifoori et al., who were able to amplify Alt a 1 gene sequences from in *A. alternata* [78] and other related taxa [79,80]. All these authors agree that the Alt a 1 sequence is an excellent tool to define a monophyletic *Alternaria-Nimbya-Embellisia-Ulocladium* and other clade with *Stemphylium* belonging to clade a very close to the mentioned above. *A. alternata* and *A. tenuissima* are taxonomically closely related species and are placed in a different group from those including *A. infectoria, U. botrytis, and S. botryosum.* Other analyzed *Pleosporaceae*, as *Curvularia lunata* or *Drechslera tritici-repentis* did not revealed homologous genes to Alt a 1 [77]. Despite the differences in Alt a 1 gene sequences among the different *Pleosporaceae* species studied, the recognition of Alt a 1 homologs by antibodies is always relevant and Alt a 1 can be considered an excellent tool to be used in the serological diagnosis of allergic diseases caused by *Alternaria.*

If we accept that the gene encoding Alt a 1-homologous-proteins is a taxonomic marker [4] and that Alt a 1, expressed in some *Pleosporaceae* species, defines a protein family [74], it would be possible that Alt a 1 would have an important biological role in the evolutionary adaptation of the *Pleosporaceae* family because Alt a 1 defines a group of phylogenetically related species.

Unfortunately, so far, no data on the biological role of Alt a 1 has been conclusively elucidated.

From an allergological point of view, it could be suggested that the Alt a 1 protein, both in its native and recombinant form, is an excellent diagnostic marker to replace the *A. alternata* extract, whether used in skin tests or for the detection and quantification of specific IgE/IgG [22]. The Alt a 1 allergen, fundamentally in its native form, is currently the most accurate and effective tool for the immunotherapeutic treatment of *Alternaria* respiratory allergy [68].

*Alternaria* species is a common saprophyte found on many plants and other substrates worldwide. It is an opportunistic pathogen that infects many agricultural crops in the field and during postharvest storage of vegetables and fruits. *Alternaria* species have been isolated from a wide range of fruits and vegetables. Certainly, this group includes the main pathogenic fungi in agriculture and food industry, resulting in severe agricultural and economic losses [81]. Accurate identification of plant pathogens is essential for understanding epidemiology and the identification of new tools for better management of the plant pathologies and postharvest contamination [82,83].

The development of DNA technology has provided effective methods for the study of different fungi, and several genetic markers have been provided for their identification. These markers are widely used in PCR or chip detection [84]. In recent years, multigene phylogeny has been widely employed for the identification and characterization of *Alternaria* species. Molecular approaches based on barcoding the gene region or gene fragments, such as the internal transcribed spacer (ITS) [85,86,87,88], mitochondrial small subunit (mtSSU), large subunit ribosomal DNA (LSU) [87], *A. alternata* major allergen (Alt a 1) [87,89,90], glyceraldehyde-3-phosphate dehydrogenase (GAPDH) [86,87,88,89], anonymous genomics regions (OPA 1–3 and OPA 2–1) [85,88,89], translation elongation factor 1 (TEF1) [87,88,89], RNA polymerase, the second largest subunit (RPB2) [87,88], plasma membrane, ATPase, calmodulin [87,89] and actin [89], have been used to define the monophyly of *Alternaria-Nimbya-Embellisia-Ulocladium* in the Ascomycete family *Pleosporaceae* relationships [74]. Current advances, especially in multi-gene phylogeny and comparative genomics, have made it possible to redefine and delineate the different *Alternaria* sections, with accurate molecular differentiation and identification of isolates showing that the *Alternaria* section consists of 11 phylogenetic species and 1 species complex [71,72,73].

Recently, some fungal allergens (Alt a 1 and Asp n 3) have been presented as valuable molecular markers of taxonomy and pathology/contamination in vegetables and fruits and also as molecular markers of allergenic contamination in indoor environments [4,84,91].

Gabriel et al., using primers defining the Alt a 1 encoding gen (390 bp) and the conserved region of Alt a 1 encoding gen (180 bp) from *A. alternata*, were able to detect the infection of citrus fruits by *A. alternata* at the onset of infection [84]. The conserved region was able to detect Alt a 1 homologs in several species of *Alternaria* section, but the fragment encoding of the Alt a 1 gen (390 bp) only detected *A. alternata* and *A. tenuissima*. They also suggest that this protein could play a role in the pathogenicity and virulence of *Alternaria* species [77,91]. Garrido-Arandia et al. propose that Alt a 1 would block some plant defense mechanisms, acting as a pathogenicity factor facilitating infection by the *Alternaria* species of the plant [92].

## 6. Conclusions

Alt a 1 defines the respiratory allergy caused by *Alternaria*, a phylogenetic related species belonging to *Pleosporaceae* family. Although the *Alternaria* taxonomy has benefited from recent phylogenetic revisions, the basis of differentiation among the major phylogenetic clades of the group is not yet understood [10,12,71,74].

It could be accepted that Alt a 1 is a protein associated with pathological phenomena in both animals and plants. Alt a 1 is useful for detecting and identifying indoor and outdoor amounts of the most important fungal allergen causing respiratory allergy and sensitization. It is tacitly accepted that Alt a 1 is an important marker for assessing the risk factor and severity of allergic respiratory disease [55]. While many efforts have been made to discover the biological role of Alt a 1, no conclusive results have been reached [70,74,92,93].

## Figures and Tables

**Table 1 jof-08-00277-t001:** Alternaria alternata allergens and their cross-reactivity with homologous proteins from other fungal species.

Allergen ^1^	Protein Type	MW (kDa)	Relevance	Homologs Cross-Reactive Allergens (Structural Database of Allergenic Proteins E Score) IUIS/Allergen.org/Allergome.org Data Bases
Alt a 1	It has a unique, dimeric β-barrel structure that define a new protein family with unknown function found exclusively in fungi.	16.4 and 15.3 bands	Major allergen	Alt b 1 *Alternaria brasicola* and other 138 *Alternaria* species Ulo c 1 *Ulocladium chartarum* Emb a 1 *Embellisia allii* and other 6 *Embellisia* species Nim c 1 *Nimbya celosiae* and other 4 *Nimbya* species Sin fu 1 *Sinomyces fusoideus* Ste b 1 *Stemphylium botryosum* and other 2 *Stemphylium* species Ulo c 1 *Ulocladium chartarum* and other 11 *Ulocladium* species
Alt a 3	Heat shock protein 70	85	Minor allergen	Pen c 19 *Penicillium citrinum* (1.9 × 10^−27^) Mala s 10 *Malassezia sympodialis* (1.2 × 10^−3^)
Alt a 4	Disulfide isomerase	57	Minor allergen	
Alt a 5	Ribosomal protein P2	11	Minor allergen	Fus c 1 *Fusarium culmorum* (3.4 × 10^−27^) Cla h 5 *Cladosporium herbarum* (8.4 × 10^−25^) Asp f 8 *Aspergillus fumigatus* (2.0 × 10^−22^)
Alt a 6	Enolase	45	Minor allergen	Cla h 6 *Cladosporium herbarum* (1.2 × 10^−157^) Asp f 22 *Aspergillus fumigatus* (1.9 × 10^−154^) Pen c 22 *Penicillium citrinum* (4.7 × 10^−153^) Cur l 2 *Curvularia lunata* (8.4 × 10^−153^) Rho m 1 *Rhodotorula mucilaginosa* (2.5 × 10^−129^)
Alt a 7	Flavodoxin, YCP4 protein	22	Minor allergen	Cla h 7 *Cladosporium herbarum* (9.4 × 10^−61^)
Alt a 8	Mannitol dehydrogenase	29	Minor allergen	Cla h 8 *Cladosporium herbarum* (6.6 × 10^−91^)
Alt a10	Aldehyde dehydrogenase	53	Minor allergen	Cla h 10 *Cladosporium herbarum* (5.9 × 10^−168^)
Alt a 12	Aid ribosomal protein P1	11	Minor allergen	Cla h 12 *Cladosporium herbarum* (1.8 × 10^−30^) Pen cr 26 *Penicillium crustosum* (4.2 × 10^−28^) Pen b 26 *Penicillium brevicompactum* (6.8 × 10^−28^)
Alt a 13	Glutathione-transferase	26	Minor allergen	
Alt a 14	Manganese SO dismutase	24	Minor allergen	Asp f 6 *Aspergillus fumigatus* (5.5 × 10^−48^) Mala s 11 *Malassezia sympodialis* (4.9 × 10^−36^)
Alt a 15	Vacuolar serine protease	58	Minor allergen	Cur l 4 *Curvularia lunata* (4.3 × 10^−152^) Cla h 9 *Cladosporium herbarum* (5.3 × 10^−119^) Pen o 18 *Penicillium oxalicum* (2.4 × 10^−114^) Asp f 18 *Aspergillus fumigatus* (4.8 × 10^−111^) Pen ch 18 *Penicillium chrysogenum* (7.9 × 10^−111^) Cla c 9 *Cladosporium cladosporioides* (1.2 × 10^−99^) Rho m 2 *Rhodotorula mucilaginosa* (1.6 × 10^−73^) Tri r 2 *Trichophyton rubrum* (1.6 × 10^−37^) Pen ch 13 *Penicillium chrysogenum* (1.1 × 10^−23^) Asp v 13 *Aspergillus versicolor* (1.5 × 10^−20^) Asp f 13 *Aspergillus fumigatus* (2.8 × 10^−20^) Asp o 13 *Aspergillus oryzae* (9.8 × 10^−19^) Asp fl 13 *Aspergillus flavus* (9.8 × 10^−19^) Pen c 13 *Penicillium citrinum* (5.3 × 10^−15^)

^1^ World Health Organization and International Union of Immunological Societies (WHO/IUIS) Allergen Nomenclature Sub-Committee Structural Database of Allergenic Proteins (SDAP), International Union of Immunological Societies. Five other *A. alternata* allergen proteins (Alt a TCTP, Alt a NTF2, Alt a 2, Alt a 9 and Alt a 70 kDa) are not included in the referred official allergen list, but they were already included in the Allergome database.

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
