# Peer review of "Fungal Allergen and Mold Allergy Diagnosis: Role and Relevance of *Alternaria alternata* Alt a 1 Protein Family"

_jof, 2022, doi:10.3390/jof8030277_

Round 1

Reviewer 1 Report

A concise, clear and objetive review of the major allergen of Alternaria was performed. There is an overall benefit to publishing this work.

Author Response

Thank you very much for your positive feedback

Reviewer 2 Report

In this manuscript the authors gave a comprehensive overview of the role of Alt a 1 protein family which is recognised gene marker in phylogeny of the species from the genus Alternaria, as well as etiological  marker for allergy and phytopathology caused by Alternaria sp. After some minor corrections I listed below, I strongly support publishing of this article.

Line 42: space prior the reference 6 is missing

Line 60: The GA2LEN initiative- since mentioned for the first time it should be explained. i.e. what is it, what is its purpose, what does it stand for 

In Table 1: "biological" should be rephrased to something more clear, e.g. protein type. In addition, the table would look more neat if the contents were positioned left 

Line 102: Rephrase the sentence- Moreover, A. fumigatus...

In the text m3 should be m3

Line 193: explain ABPA

Line 318: Fullstop should be placed after, and not before the reference 56.

The last sentence of the conclusion should be rephrased. 

Author Response

Thank you very much for your comments.

Line 42: space prior the reference 6 is missing-mira a ver si ha cambiado el número de línea con las modificaciones y si es así pones:

Line 42, now line XX

In the text m3 should be m3

Line 318: Fullstop should be placed after, and not before the reference 56.

We have made changes to the format errors (marked using the “Track Changes” function of MSWord)

Line 60: The GA2LEN initiative- since mentioned for the first time it should be explained. i.e. what is it, what is its purpose, what does it stand for 

We have made a brief explanation on GA2LEN Project

Line 193: explain ABPA

The abbreviation ABPA was previously described in the manuscript on line 98 (new version).

In Table 1: "biological" should be rephrased to something more clear, e.g. protein type. In addition, the table would look more neat if the contents were positioned left 

Line 102: Rephrase the sentence- Moreover, A. fumigatus...

The last sentence of the conclusion should be rephrased. 

Thank you for your comments. We have made changes in the table and sentences according to your suggestions.

Reviewer 3 Report

The review “Alt a 1 protein family: Phylogenetic-related Alternaria marker for taxonomy, allergy and Phytopathology by Sanchez and collaborators, is focused mainly on the role of the Alternaria a.and other fungi or molds extracts and the molecules in allergy diagnosis.

However the title chosen by the authors is irrelevant to identify and define the content of the manuscript. The review is focused on the fungal allergens and mold allergy diagnosis, on allergological relevance of the mold distribution and the use of fungal extracts in allergy diagnosis, on the application of fungal molecules in molecular diagnosis with an in-depht study on Alternaria a.molecules. Therefore I suggest the authors to adapt the title of the manuscript to the content or the other way around.

Minor points

Lines 221-222. The title of this section corresponds to the title of the review. Please change it.

Lines 261-262.  A homologous protein of Alt a 1 are is expressed in some species of….Please change this sentence.

Author Response

Thank you very much for your comments.

We have changed the title to adapt it to the content of the manuscript

Minor points

Lines 221-222. The title of this section corresponds to the title of the review. Please change it.

We have kept the title of the section with minor changes since the general title of the manuscript has been modified.

Lines 261-262.  A homologous protein of Alt a 1 are is expressed in some species of….Please change this sentence.

We have rephrased the sentence as follows in line 265-269:

“If we accept that the gene encoding Alt a 1-homologous-proteins is a taxonomic marker [4] and that Alt a 1, expressed in some Pleosporaceae species, defines a protein family [75], it would be possible that Alt a 1 would have an important biological role in the evolutionary adaptation of Pleosporaceae family because Alt a 1 defines a group of phylogenetically related species.”

Round 2

Reviewer 3 Report

The authors took into account the changes requested by the reviewer. The manuscript can be accepted in this form.

Best regards